# FMISO-Based Adaptive Radiotherapy in Head and Neck Cancer

**DOI:** 10.3390/jpm12081245

**Published:** 2022-07-29

**Authors:** Martin Dolezel, Marek Slavik, Tomas Blazek, Tomas Kazda, Pavel Koranda, Lucia Veverkova, Petr Burkon, Jakub Cvek

**Affiliations:** 1Department of Oncology, Palacky University Medical School & Teaching Hospital, 77900 Olomouc, Czech Republic; martin.dolezel@fnol.cz; 2Department of Radiation Oncology, Masaryk Memorial Cancer Institute, 65652 Brno, Czech Republic; tomas.kazda@mou.cz (T.K.); burkon@mou.cz (P.B.); 3Department of Radiation Oncology, Faculty of Medicine, Masaryk University, 62500 Brno, Czech Republic; 4Department of Oncology, Faculty of Medicine, University Hospital Ostrava, 70852 Ostrava, Czech Republic; tomas.blazek@fno.cz (T.B.); jakub.cvek@fno.cz (J.C.); 5Department of Nuclear Medicine, Palacky University Medical School & Teaching Hospital, 77900 Olomouc, Czech Republic; pavel.koranda@fnol.cz; 6Department of Radiology, Palacky University Medical School & Teaching Hospital, 77900 Olomouc, Czech Republic; lucia.veverkova@fnol.cz

**Keywords:** head and neck cancer, adaptive radiotherapy, FMISO

## Abstract

Concurrent chemoradiotherapy represents one of the most used strategies in the curative treatment of patients with head and neck (HNC) cancer. Locoregional failure is the predominant recurrence pattern. Tumor hypoxia belongs to the main cause of treatment failure. Positron emission tomography (PET) using hypoxia radiotracers has been studied extensively and has proven its feasibility and reproducibility to detect tumor hypoxia. A number of studies confirmed that the uptake of FMISO in the recurrent region is significantly higher than that in the non-recurrent region. The escalation of dose to hypoxic tumors may improve outcomes. The technical feasibility of optimizing radiotherapeutic plans has been well documented. To define the hypoxic tumour volume, there are two main approaches: dose painting by contour (DPBC) or by number (DPBN) based on PET images. Despite amazing technological advances, precision in target coverage, and surrounding tissue sparring, radiation oncology is still not considered a targeted treatment if the “one dose fits all” approach is used. Using FMISO and other hypoxia tracers may be an important step for individualizing radiation treatment and together with future radiomic principles and a possible genome-based adjusting dose, will move radiation oncology into the precise and personalized era.

## 1. Introduction

Head and neck cancers are a broad variety of malignant tumors affecting the oral cavity, head, and neck region. The most common one is squamous cell carcinoma. Although various treatments have been proposed, the gold standard therapy for the management of these lesions is surgery, followed by radiotherapy in cases of relapses or when surgery is not possible [1,2]. Radiotherapy (RT) and concurrent chemotherapy represent one of the most commonly used strategies in the curative treatment of patients with head and neck cancer (HNC). Tumor hypoxia belongs to the main cause of treatment failure in many types of cancer including HNC [3]. The significance of a lack of oxygen was demonstrated by Schwarz in the first radiobiologically oriented clinical study already in 1909. Hypoxia has been directly identified in most animal solid tumors, with the values ranging from less than 1% to well more than 50% of the total viable cell population. In the 1950s, tumor hypoxia was first described by radiation oncologists as a frequent cause of failure to radiotherapy in solid tumors [4].

Hypoxia in solid tumors is basically due to the decreased delivery of oxygenated blood to cover the metabolic demands of the rapidly proliferating tumor cells and is augmented by further pathogenic factors such as structurally and functionally abnormal tumor microvasculature with increased distances between tumor microvessels [5]. These pathological features together with tumor- and treatment-triggered anemia result in chronic hypoxia. Acute hypoxia caused by a transient reduction in perfusion may also occur in tumors. Both types of hypoxia contribute to a highly dynamic microenvironment where cells are exposed to differential oxygen gradients both spatially and temporally [5]. The changes in gene expressions are controlled by the family of heterodimeric transcription factors—hypoxia-inducible factors (HIFs)—and favor survival in a hostile environment under hypoxic conditions [6]. HIFs are involved in the expression of genes that control glucose uptake, metabolism, angiogenesis, erythropoiesis, cell proliferation, apoptosis, metastasis, and thus regulate multiple aspects of tumorigenesis and the response to radiation therapy [4]. There is some evidence, that the HIF-1—the first HIF family member and critical regulator of the cellular response to hypoxia—promotes radioresistance and the HIF-1-deficient tumors are more sensitive to radiation compared to wild-type tumors [4]. 

Tumor hypoxia is often associated with tumor necrosis and usually reflects the imbalance between tumor growth and the vascular supply required for oxygen and nutrient delivery. In general, a fluid-containing metastatic node is defined as necrotic, and the incidence of lymph nodal necrosis is present in 44.0% of advanced head and neck cancer [7]. Necrosis could be identified by modern imaging methods and was considered an independent prognostic factor for nasopharyngeal cancer treated by radiotherapy [8]. Liang et al. reported that 41% and 55% of patients with and without necrosis of the total tumor achieved a complete response, respectively [9]. Similarly, Ou et al. demonstrated that hypoxia-related biomarkers were associated with poor local control in p16-negative tumors [10]. Nevertheless, hypoxia plays a significant role in HPV-related tumors. Head and neck squamous cell carcinomas are characterized by significant genomic instability that could lead to clonal diversity due to the random cellular accumulation of mutations. A biopsy might not be representative of the heterogeneity of hypoxia within a whole tumour; Zhang et al. characterized the high degree of intratumor genetic heterogeneity within a single tumor based on the whole-genome sequencing on three separate regions of HPV-positive oropharyngeal squamous cell carcinomas [11]. Another important tumor characteristic is the phenomenon of epithelial–mesenchymal transition (EMT) contributing to metastasis [12]. Several molecular mechanisms have been identified as inducers of EMT in cancer cells. Hypoxia, through the actions of HIF-1α, plays a significant role in this process that leads to metastasis due to the loss of cell adhesion and increased cell motility [13]. Although the contribution of EMT to radioresistance in vivo remains unexplored, Johansson et al. demonstrated that EMT is associated with a poor radioresponse in vitro [14].

Furthermore, in the presence of molecular oxygen at the time of exposure, low-linear energy transfer (LET) radiation ionizes water molecules, producing high-energy electrons and highly reactive chemicals formed from O2 (highly reactive oxygen species—ROS). DNA damage results from either a direct or an indirect (via highly reactive oxygen species) effect of irradiation. In the absence of oxygen, highly reactive oxygen species are not produced, and DNA damage is reduced.

Estimating hypoxia in human tumors has generally involved the use of indirect methods. The hypoxia features included immunohistochemical estimates of intercapillary distance, vascular density, and distance from tumor cells to the nearest blood vessel; oxyhemoglobin saturation determined using cryophotometry, tissue protein expression analyses, or noninvasive examinations with magnetic resonance imaging (MRI); measurements of tumor perfusion using MRI, computed tomography (CT), or positron emission tomography (PET) using radioactively labeled nitroimidazoles (18F labeled misonidazole, 123I labeled azomycin arabinoside) [3,15,16,17,18,19,20].

The relationship between pre-treatment measurements of tumor oxygen tension (pO2) and survival in advanced head and neck cancer was proven by Nordsmark in 2005 [21]. The second Danish Head and Neck Cancer Study (DAHANCA 2) found a highly significant improvement in the stratification subgroup of pharynx tumors using misonidasole [22]. The recent meta-analysis of randomized clinical studies in squamous cell carcinoma of the head and neck using hypoxic radiosensitizers to improve radiotherapy clearly showed that radiosensitizer modification of tumor hypoxia significantly improved locoregional tumor control and overall survival, with odds ratios of 0.71 and 0.87, respectively [23]. Besides radiosensitizers, optimized RT overcoming tumor hypoxia by safe dose escalation may improve the therapeutic ratio and some HNC patients at a high risk of treatment failure may benefit from this approach [23].

## 2. Hypoxia in Head and Neck Cancers

### 2.1. Hypoxia Evaluation Using FMISO

Positron emission tomography (PET) using hypoxia radiotracers has been studied extensively and has proved its feasibility and reproducibility to detect tumor hypoxia. PET tracers containing the oxygen-sensitive nitroimidazole group are specifically designed to detect hypoxic regions [24,25]. The first specific hypoxia PET tracer was 18F-labeled fluoromisonidazole (18F-FMISO), and it is currently the most frequently used and studied tracer for this purpose [26,27]. More hypoxia-specific tracers studied in HNC are 18F-fluoroazomycin arabinoside (FAZA), 18F-EF5 (18F-2-nitroimidazole-pentafluoropropyl acetamide), 18F-EF3 (18F-2-nitroimidazol-trifluoropropyl acetamide), and a few others [28]. FAZA was developed to avoid the drawbacks of the high lipophilicity of FMISO although not all comparative preclinical studies showed a clear superiority of this novel tracers [29].

The mechanism of 18F-MISO accumulation has been described by Padhani in detail [30]. The partition coefficient of 18F-MISO (Figure 1) nearly equals one, so the molecule freely diffuses into all cells. Once 18F-MISO is in an environment with electron transport occurring, the –NO2 substituent takes on an electron to form the radical anion reduction product. If O2 is also present, that electron is rapidly transferred to oxygen, and 18F-MISO changes back to its original structure and leaves the cell. However, if a second electron from cellular metabolism reacted with the nitroimidazole to form the two-electron reduction product, the molecule reacts non-discriminately with peptides and RNA within the cell and becomes trapped. Thus, the retention of 18F-MISO is inversely related to the intracellular partial pressure of O_2_. It has been reported that 18F-FMISO-PET reflects cell reoxygenation, is highly reproducible, and in HNC patients’ population showed a stable conformation of the hypoxic subvolumes during chemoradiation therapy, and therefore seems to be appropriate for monitoring therapeutic efficiency [25,26,27]. Due to the low signal-to-noise ratio of hypoxia-specific PET tracers, an optimal reference region of interest (ROI) is of utmost importance. The probably most relevant meta-analysis in the clinic validated the use of a deep neck muscle ROI as the most appropriate and reliable background ROI for HNC patients since these muscles are always in the field of view. However, the use of other background ROIs, e.g., cerebellum, carotic arteries, left ventricle, or aorta is also possible [29]. Discriminating the hypoxic volume (HV), the most utilized thresholds of 1.2 and above in case of tumor to blood ratio, and utilization of similar thresholds is just in case of a tumor to muscle ratio of 1.2, 1.4, and above with the strongest correlation with locoregional control proven for HV1.6. [29,31].

### 2.2. Locoregional Relapse

Historically, radiation therapy alone was the standard nonsurgical therapy for locally advanced disease. Unfortunately, radiotherapy regimens result in local control rates of only 50% to 70% and disease-free survivals (DFSs) of 30% to 40% [32]. This meta-analysis of individual patient data from >17,346 participants in 93 trials conducted from 1965 to 2000 (Meta-Analysis of Chemotherapy in Head and Neck Cancer MACH-NC) demonstrated that the use of radiotherapy and concurrent chemotherapy resulted in a 19% reduction in the risk of death and an overall 6.5% improvement in 5-year survival compared to treatment with RT alone [32]. This benefit was predominantly attributable to a 13.5% improvement in locoregional control [32]. 

Thus, RT and concurrent chemotherapy represent the most used strategy and is a biologically attractive approach because some chemotherapeutic agents may both radiosensitize cells and provide additive cytotoxicity.

Locoregional failure is the predominant recurrence pattern, and most deaths from HNC are due to uncontrolled local and/or regional disease. Hypoxia occurs in approximately 80% of head and neck tumors [21]. Based on experimental and clinical data, hypoxia is a useful parameter for pretherapeutic stratification. Besides tumor volume correlating with cell number per tumor, hypoxia is an important biological parameter for tumor progression. Hypoxia increases radioresistance and is a predictive factor for local failure, based on retrospective data suggesting that loco-regional recurrences after chemoradiotherapy originate from the initial GTV containing hypoxic subvolumes [21]. Nishikawa confirmed that the uptake of FMISO in the recurrent region is significantly higher than that in the non-recurrent region [33]. Similarly, Carles et al. demonstrated less probable recurrence in HNC patients with an increase in the FMISO heterogeneity (increasing low concentration regions) probably by an improvement in tumor cell re-oxygenation during the course of chemoradiation [25]. The large multicenter meta-analysis of individual patient data proved PET-measured hypoxia is robust and has a strong impact on LRC and OS in HNC [29].

Thus, the optimization of radiotherapy in hypoxic subvolumes represents one of the most important unmet clinical needs in HNC patients. Dose painting by numbers can be an elegant way to individualize radiotherapy by functional imaging such as PET, which could overcome the resistance of hypoxic HN spinocellular carcinomas. 

### 2.3. Adaptive HNC Radiotherapy

Anatomical changes during the course of treatment can also have a significant impact on treatment outcome. Patients may therefore benefit from the implementation of both biological and anatomical image-guided adaptive radiotherapy. Anatomic changes in the patient due to weight loss, inflammation, edema, muscle atrophy, and/or tumor reduction during radiotherapy schedule have a significant impact on the delivered dose. This issue is even more imperative under conditions of possible dose escalation. Bhandari et al. reported a 10% weight loss after the third week of radiation therapy [34]. Bhide et al. prospectively assessed changes in treated volumes using a weekly CT and described a reduction in the clinical target volume at the level of 10.5% between week 0 and week 2 of radiotherapy and a 15% reduction in the parotid volumes by week 2 and 31% by week 4 [35]. Adaptive radiotherapy refers to acquiring a new set of imaging followed by a process of re-planning patients in predefined circumstances. As a result, adaptive radiotherapy has the potential to decrease toxicity and improve local control for locally advanced HNC. There are not many studies published on this issue so far. In a prospective study from China published in 2013, eighty-six patients with advanced nasopharyngeal carcinoma underwent routine CT-based re-planning, mostly before the 25th fraction, while 43 patients refused a repeat CT scan, did not have re-planning, and served as a control group. Two-year locoregional control was better in the re-planning group (97% vs. 92%, *p* = 0.04) as well as the quality of life being improved [36]. Better tissue sparing was shown in another prospective study with 22 oropharyngeal patients who all had re-planning on a weekly basis [37]. Improved normal dose and target coverage and better 3-years local relapse free survival in T3 and T4 tumors were demonstrated in a retrospective study with 33 out of 66 nasopharyngeal patients re-planned, while improved local control was proven in a retrospective study with 51 re-planned patients out of 317 various HNC patients [38,39].

Nevertheless, identifying patients who may significantly benefit from adaptive RT prior to the start of their radiotherapy course is difficult [40]. There are also considerable differences in the timing of adaptation across studies. The timing of adaptation is usually under the discretion of the treating physician. A timepoint of around 15–20 fractions of radiotherapy was selected by most authors [41]. Some authors tried to objectivize the ideal time point for adaptation. One group suggests that an adaptive re-plan can be triggered using spinal cord doses calculated on the CBCT. Implementing this trigger can reduce patient appointments and the workload of all personnel by eliminating up to 90% of additional unnecessary CT scans, and the authors further argue that there is evidence that dosimetric increases push the OAR dose above tolerance in only a small minority of cases [42]. But this might be contradictory keeping in mind the differences in treatment volumes dose distribution in the above-mentioned studies and when the volume of parotids reduced with further possibility of hampering their tolerance with the quality-of-life consequences [43]. On the other hand, Hunter et al. concluded that replanning is unlikely to improve salivary output after treatment in most cases even though replanning can reduce the mean dose to the parotids basically because of the importance of the remaining function of the other salivary glands [44]. Stauch et al. investigated the dosimetric impact of weight loss and anatomical separation difference in head and neck (H&N) patients and examined the effectiveness of adaptive planning and compared VMAT and IMRT adaptive plans [45]. Despite some differences, were found: the mean weight loss was 9% and the mean anatomical difference measured in nine vectors in levels of C1, C3, and C4/5 was 1.06 cm, the coverage of all targets improved on average regarding both VMAT and IMRT. Specifically, D95 of the new planning target volume with the highest prescribed dose increased 0.77% and 0.60% for VMAT and IMRT plans, respectively, and a mean increase of 1.25% and 1.01% was found for D95 of the clinical target volume. Most risk structures also received an additional dose with the largest increase for the pharynx. The spinal cord received a mean increase of 1.8 Gy and 1.5 Gy for the VMAT and IMRT plans, respectively. Mean dose of parotids increased 4.1 Gy and 3.9 Gy for the VMAT and IMRT plans, respectively. No quantitative method for finding the threshold of anatomical separation difference requiring a replanning was established. Since weight loss is a gradual process and the true overdosing is even smaller than in the time of replanning and the changes in organ dose were marginal, authors conclude that adaptive radiotherapy may not always be necessary when the alignment of bony anatomy and remaining soft tissue is within tolerance and physician judgment and preference is still needed [45]. Despite the various facts concerning adaptation, its importance stands out even more in the case of further dose intensification.

### 2.4. Dose-Escalated HNC Radiotherapy

International guidelines recommend a dose between 60 and 70 Gy. Many studies have explored a wide range of altered fractionation schedules as modern, highly conformal techniques that extend the dose range while maintaining organs at risk constraints. Doses to the tumour volume of 1.5 Gy up to 2.5 Gy per fraction were prescribed and total doses ranged from 74 Gy in 2.42 Gy daily fractions (EQD2 76.59 Gy10) to 75 Gy in 2.5 Gy fractions (EQD2 78.13 Gy10). The highest equivalent dose delivered was a hyperfractionated, accelerated regimen that gave 2 Gy for the first 10 fractions then 1.8 Gy twice daily for the remaining 15 fractions to a total of 74 Gy (EQD2 81.2 Gy10) [46]. However, it is still an open question whether dose-escalation is a reasonable approach to improve locoregional control. On the other hand, adding an additional radiation dose to standard schedules seems to be more cost-effective than alternatives such as modern systemic therapy [46]. Using standard anatomical imaging to guide target delineation, a dose of 70.8 Gy in 30 fractions of 2.36 Gy was defined as the maximum tolerated dose deliverable to the GTV using this accelerated fractionation with simultaneous integrated boost intensity-modulated radiotherapy regimen [47]. Dose escalation seems to be feasible to at least 10 Gy10 above the conventionally recommended dose and fractionation [47]. Later, the Japanese authors increased the dose on involved nodes with a diameter above 2 cm by the intentionally internal high-dose (IIHD) policy. The IIHD area was contoured inside the lymph node and delivered 110% to 150% of the prescription dose. When substantial shrinkage of the tumor or body shape occurred, re-planning was performed immediately to make an adaptive treatment plan. When the lymph node metastases had shrunk to less than 2 cm in diameter under the IIHD treatment, the IIHD areas were erased. The median IIHD volume was 1.7 cm^3^ (range: 0.1–76.5 cm^3^), and the median percentages of the volumes actually irradiated 110% dose to GTVn volumes were 41.2% (range: 12.5–92.0%). By maintaining PTVn, D98% did not significantly increase (median, 101.3%) with no OAR increased dose and there was no difference in grade 2/3 dermatitis and mucositis in contrast to previously published studies. Locoregional relapse-free survival was significantly longer in the IIHD group. This approach is deemed feasible and very similar to those with contouring F-FMISO hypoxic subvolumes [48].

### 2.5. FMISO-Based Adaptive Radiotherapy

Currently, in daily practice, an identical radiation dose is delivered to all subvolumes of the tumor regardless of individual biology or radiosensitivity. The concept of “dose painting” involves adapting the dose prescriptions for different tumor subvolumes according the cancer heterogeneous biology. This could be done using functional imaging that shows hypoxic subvolumes of tumors resulting in a “biological target volume” (BTV). To counteract radioresistance associated with hypoxic tumors, radiation oncologists can escalate the dose to these hypoxic regions of cancer to achieve better tumor control without compromising normal tissue tolerance [49,50].

The escalation of dose to hypoxic tumors may (in theory) improve outcomes. The technical feasibility of optimizing radiotherapeutic plans has been well documented, mostly in head and neck cancers. To define the hypoxic tumour volume, there are two main approaches: dose painting by contour (DPBC) or by number (DPBN) based on PET images [51].

Several studies are currently underway to focus on the use of FMISO in the adaptation of the irradiation protocol (Table 1). The dose escalation ranges from 77 to 80.5 Gy throughout these studies. In the German Escalox trial, the planned accrual is 250 patients, and in two experimental arms the escalated dose comprises whole GTV apart from the proximity of critical structures, where the 0.3 mm separation margin is applied [49]. The assessment of FMISO-related hypoxia will be a secondary objective and for this reason 100 patients will undergo FMISO PET/CT twice a week before the initiation of radiation therapy.

The planned Czech study Farhead (NCT05348486) is more “FMISO guided” and will elevate the dose to the hypoxic regions on the pre-treatment FMISO scan and modify (also elevate) the dose in accordance to intra-treatment FMISO carried out after the 11th fraction. The dose elevation will provide all 60 planned patients and the primary endpoint is the feasibility and survival parameters that will be compared to historical cohorts.

So far, there is only one randomized phase II study (NCT02352792) recently published by Welz et al. on this issue with 53 locally advanced HNC patients enrolled; out of them, 39 (74%) had hypoxic tumors [50]. Patients were randomized irrespective of hypoxia into standard treatment (70 Gy/35 fractions) or dose-escalated arm (77 Gy/35 fractions, by 2.2 Gy per fraction as simultaneous boost). For non-hypoxic patients, 100% 5-year LC (local control) was observed compared to 74% in patients with hypoxic tumors (*p* = 0.039). However, the hypothesis that a dose escalation of 10% to a HV was able to overcome hypoxia-induced resistance could not be confirmed (*p* = 0.150), although a 25% higher 5-year LC in the dose-escalated arm compared to the standard treatment arm may support the dose escalation concept. Moreover, this study closed prematurely due to slow accrual and an insufficient number of patients. There are also concerns that the increased dose was lower—only 2% than the intended 10% (to 77 Gy), and thus inadequate because of the small size of the HV. RT planning constrains and no margins on HV led to the small absolute dose-escalated volumes.

The previously published largest metanalysis of individual data from 153 patients found the strong correlation between hypoxic subvolume and LRC and OS, but unfortunately, with no effect of augmented treatment, most likely due to low number of patients, only 10 out of 153 patients underwent dose escalation so it is fairly reliable [29].

The presence of hypoxia in the tumor can be utilized by two basic principles. The first is to identify a group of low-risk patients and de-escalate the radiotherapy regimens, while the second is to specify a high-risk cohort and intensify the treatment (Figure 1).

Examples of low-risk patients are patients with proven HPV positivity without the presence of hypoxia. A large number of phase II dose reduction studies are currently underway in HPV-positive patients due to the excellent cancer outcomes using standard therapy. Dose de-escalation could lead to equivalent treatment efficacy and reduced toxicity. The use of FMISO in this group of patients would theoretically further specify and objectify the target group. An example is the ongoing study NCT03323463, which reduces the dose to 30 Gy in this cohort [52].

The opposite example are HPV-negative patients with hypoxia. In them, theoretically, the escalation of therapy could lead to better locoregional control. This procedure is used in an above-mentioned randomized phase III ESCALOX study in which patients in the dose-modified arm escalated to 80.5 Gy [49].

## 3. Conclusions

Despite amazing technological advances, precision in target coverage and surrounding tissue sparring, radiation oncology is still not considered a targeted treatment as long as the “one dose fits all” approach is used. Using FMISO and other hypoxia tracers might be an important step for individualizing radiation treatment and together with future radiomic principles and a possible genome-based adjusting dose will move radiation oncology into the precise and personalized era.

## Figures and Tables

**Figure 1 jpm-12-01245-f001:**
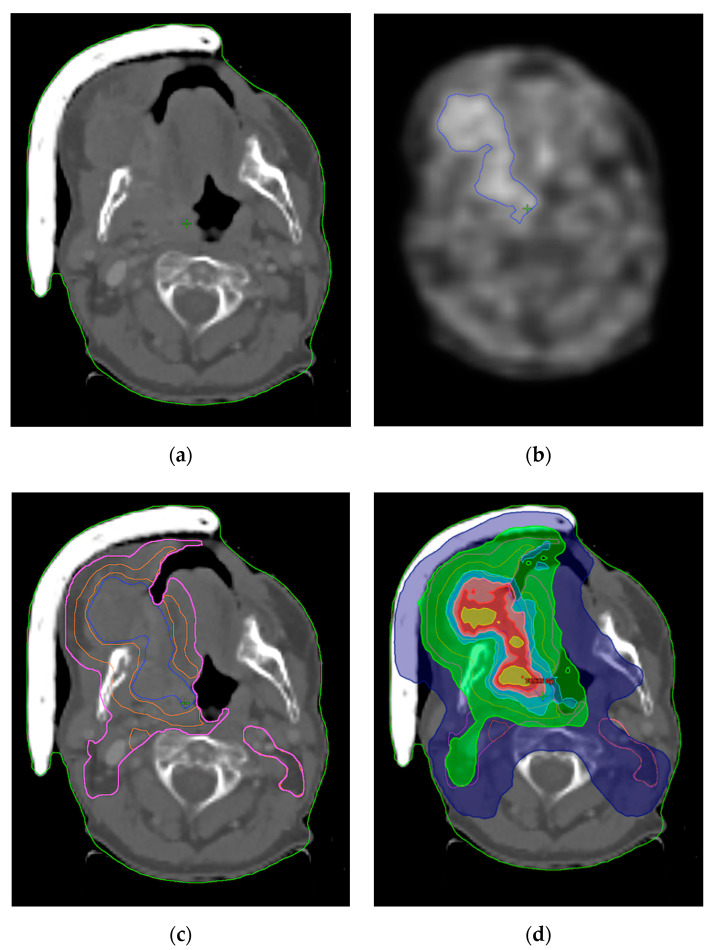
Dose escalation in patient with cancer of oral cavity using FMISO. (**a**) Planning CT; (**b**) PET/CT using FMISO with hypoxic region (blue contour); (**c**) Planning CT with contours; (hypoxic region-blue, 70 Gy isodose-orange, 50 Gy isodose-pink; (**d**) Dose distribution with escalation in the hypoxic region. Dark blue-60% of the prescribed dose, 42 Gy, Green-95% of the prescribed dose, 66.5 Gy, Cyan-103% of the prescribed dose, 72.1 Gy Pink-105% of the prescribed dose, 73.5 Gy, Red-108.5% of the prescribed dose, 75.9 Gy Yellow-110% of the prescribed dose, 77 Gy.

**Table 1 jpm-12-01245-t001:** Ongoing studies using FMISO in the adaptation of the irradiation protocol.

Study Number	Study Original Name	Intervention	Type	Year
NCT00606294 (USA)	A Study Using Fluorine-18-Labeled Fluoro-Misonidazole Positron Emission Tomography to Detect Hypoxia in Head and Neck Cancer Patients	*Cohort 1* Primary tumor 70 Gy, positive nodes 70 Primary tumor 70 Gy, positive nodes 60 (HPV+/FMISO−) *Cohort 2* Surgical bed 30 Gy, positive nodes 30+/− neck dissection (HPV+/FMISO−)	Prospective	2008
NCT05348486 (Czech Republic)	FARHEAD: FMISO-based Adaptive Radiotherapy for Head and Neck Cancer	*Standard arm* Primary tumor and positive nodes 70 Gy/33 fractions (HPV−/FMISO−) *Experimental arm* Primary tumor and positive nodes 75.9–79.2 Gy/33 fractions (HPV−/FMISO+)	Prospective	2022
NCT03865277 (Austria)	Individualized Radiation Dose Prescription in HNSCC Based on F-MISO-PET Hypoxia-Imaging: Multi-center, Randomized Phase-II-trial	*Standard hypoxic* Primary tumor and positive nodes 70/35 Gy fractions (HPV−/FMISO+) *Experimental dose-escalated hypoxic* Primary tumor and positive nodes 77 Gy/35 fractions (HPV−/FMISO+) *Experimental dose-escalated carbon hypoxic* Primary tumor and positive nodes 77 Gy/35 fractions (HPV−/FMISO+) *Standard oxic* Primary tumor and positive nodes 70/35 fractions Gy (HPV−/FMISO−) *Standard HPV+* Primary tumor and positive nodes 70/35 fractions Gy (HPV+/FMISO+−)	Phase II	2022
NCT02352792 (Germany)	Randomized Phase II Study for Dose Escalation in Locally Advanced Head and Neck Squamous Cell Carcinomas Treated With Radiochemotherapy	*Standard arm* Primary tumor and positive nodes 70 Gy/33 fractions *Experimental arm* Standard plus 10% dose escalation to the hypoxic volume	Phase II	2015
NCT02207439 (USA)	A Phase II Trial of a Protease Inhibitor, Nelfinavir (NFV), Given With Definitive, Concurrent Chemoradiotherapy (CTRT) in Patients With Locally Advanced, Human Papilloma Virus (HPV) Negative, Squamous Cell Carcinoma of the Head and Neck	Nelfinavir for 7–14 days prior chemoradiotherapy (HPV−/FMISO+)	Phase II	2014
NCT01212354 (Germany)	Escalox—Phase III A Prospective, Randomized, Rater-blinded, Multicentre Interventional Clinical Trial. Do Selective Radiation Dose Escalation and Tumour Hypoxia Status Impact the Locoregional Tumour Control After Radiochemotherapy of Head and Neck Tumours?	*Standard arm* Primary tumor and positive nodes 70 Gy/33 fractions (5 × 2 Gy per week) *Experimental arm* Primary tumor and positive nodes 80.5 Gy/33 fractions (5 × 2.3 Gy per week)	Phase II	2010
NCT03323463 (USA)	A Prospective Single Arm Non-inferiority Trial of Major Radiation Dose De-Escalation Concurrent With Chemotherapy for Human Papilloma Virus Associated Oropharyngeal Carcinoma	*Standard arm* Primary tumor and positive nodes 70 Gy/33 fractions (HPV+/FMISO+) *Experimental arm* Primary tumor and positive nodes 30 Gy/10 fractions (HPV+/FMISO−)	Prospective	2017

## Data Availability

Not applicable.

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
