# Peer review of "FMISO-Based Adaptive Radiotherapy in Head and Neck Cancer"

_jpm, 2022, doi:10.3390/jpm12081245_

Round 1
Reviewer 1 Report
A very interesting and well-performed narrative review about the use of 18F-labeled fluoromisonidazole radiotherapy in the management of head and neck cancer, also including currently on going trials. Exploring the efficacy of hypoxia tracers will be fundamental for individualizing radiation treatment and moving radiation oncology to more personalized treatments;
only minor queries:
line 34: you should add a small description for head and neck cancer, such as: " Head and Neck Cancers are a broad variety of malignant tumors interesting the oral cavity, head, and neck region. the most common one is squamous cell carcinoma. Although various treatments have been proposed, the gold standard therapy for the management of these lesions is surgery, followed by radiotherapy in cases of relapses or when surgery is not possible." and cite: doi: 10.3390/curroncol28040213. and doi: 10.3390/medicina57060563.
Thank You
Author Response
Thank you for your revision and suggested text. It has been added to the original article in accordance with your recommendation together with suggested citations.
Reviewer 2 Report
General comments
1 The authors presented a review on adaptive radiotherapy (ART) on head and neck cancer covering mainly adaptation against biological changes during the course of radiotherapy. The manuscript is very well written on mechanism and impact of hypoxia in management of head and neck cancer and some clinical studies on PET-FMISO-based dose escalation and adaptation in IMRT and VMAT treatments. The paper should be of interest to the community.
2 Hypoxia plays a key role in the failure of radiotherapy treatment of head and neck cancer. The application of hypoxia imaging-guided adaptation can indeed improve treatment outcome. However, anatomical changes during the course of treatment can also have a significant impact on treatment outcome. FMISO-based adaptation alone may not be able to achieve the treatment objective, especially when such adaptation is applied infrequently. Patient may benefit from implementation of both anatomical and biological image-guided ART.
Specific comments
Line 96- Reference should be quoted for this statement.
Line 146-149: Optimization and adaption of hypoxic volumes by means of dose painting demands high precision target and OAR localization consistently throughout the treatment course. This cannot be achieved without appropriate corrective measures. Online anatomical image-guided adaption, which is now widely used is a suitable corrective measure which may help achieving the treatment objective.
Line 176- The application of adaptation after 15-20 fractions may be appropriate time for hypoxia adaptation. The anatomical changes during this period might have significantly affected the dose and dose distribution.
Line 178-179: Calculation of cord dose based on CBCT images is subject to challenge due to the uncertainties involved.
Line 211-213- The meaning of this sentence is not clear.
Author Response
Many thanks for reviewing our manuscript and for the interesting queries. We tried to reply all of them and the following texts and a citation were added to the manuscript.
Specific comments:
Line 96- Reference should be quoted for this statement.
Padhani AR, Krohn KA, Lewis JS, Alber M. Imaging oxygenation of human tumours. Eur Radiol 17: 861–872 2007
Line 146-149: Optimization and adaption of hypoxic volumes by means of dose painting demands high precision target and OAR localization consistently throughout the treatment course. This cannot be achieved without appropriate corrective measures. Online anatomical image-guided adaption, which is now widely used is a suitable corrective measure which may help achieving the treatment objective.
Thank you for your comment, which we completely agree with. The following clarification was made in the manuscript using your comment:
Line 150 resp Line 177 in the new version of the manuscript - Anatomical changes during the course of treatment can also have a significant impact on treatment outcome. Patients may therefore benefit from the implementation of both biological and anatomical image-guided ART.
Line 176- The application of adaptation after 15-20 fractions may be appropriate time for hypoxia adaptation. The anatomical changes during this period might have significantly affected the dose and dose distribution.
Thank you for your comment, which we completely agree with
Line 178-179: Calculation of cord dose based on CBCT images is subject to challenge due to the uncertainties involved.
Thank you for your comment, which we completely agree with
Line 211-213- The meaning of this sentence is not clear.
We deeply apologize for the mistake, the sentence has been modified as follows:
Line 238 resp. 241 in the new versionof the manuscript: Many studies have explored a wide range of altered fractionation schedules as modern, highly conformal techniques extend the dose range while maintaining organs at risk constraints.
Reviewer 3 Report
The manuscript made by Dolezel M et al is interesting updated and well done, the manuscript is focus on relationship of hypoxia and radiotherapy.
I have some questions that authors need resolve
First: Would it be possible discuss the importance the necrosis reported in histopathological studies related with RT?
Second: Would it be possible explain the tumor homogeneity and heterogeneity related with VPH and hypoxia?
Third: Would it be possible explain the EMT phenomena related to hypoxia and RT?
Author Response
Many thanks for reviewing our manuscript and for the interesting queries. We tried to reply all of them and the following texts and citations were added to the manuscript.
First: Would it be possible to discuss the importance of the necrosis reported in histopathological studies related to RT?
Line 63-72: The tumor hypoxia is often associated with tumor necrosis and usually reflects the imbalance between tumor growth and the vascular supply required for oxygen and nutrient delivery. In general, a fluid-containing metastatic node is defined as necrotic and the incidence of lymph nodal necrosis is present in 44.0% of advanced head and neck cancer (1). Necrosis could be identified by modern imaging methods and was considered an independent prognostic factor for nasopharyngeal cancer treated by radiotherapy (2). Liang et al. reported that 41% and 55% of patients with and without necrosis of the total tumor achieved complete response respectively (3). Similarly, Ou et al. demonstrated that hypoxia-related biomarkers were associated with poor local control in p16-negative tumors (4).
Second: Would it be possible explain the tumor homogeneity and heterogeneity related with VPH and hypoxia?
Head and neck squamous cell carcinomas are characterized by significant genomic instability that could lead to clonal diversity due to random cellular accumulation of mutations. A biopsy might not be representative of the heterogeneity of hypoxia within a whole tumour, Zhang et al. characterized the high degree of intratumor genetic heterogeneity within a single tumor based on the whole-genome sequencing on three separate regions of HPV-positive oropharyngeal squamous cell carcinoma (5).
Third: Would it be possible explain the EMT phenomena related to hypoxia and RT?
Epithelial-mesenchymal transition (EMT) is an important phenomenon contributing to metastasis. Several molecular mechanisms have been identified as inducers of EMT in cancer cells. Hypoxia, through the actions of HIF-1α, plays a significant role in this process that lead to metastasis due to the loss of cell adhesion and increased cell motility (6). Although the contribution of EMT to radioresistance in vivo remains unexplored, Johansson et al. demonstrated that EMT is associated with a poor radioresponse in vitro (7).
- Lan, M., Huang, Y., Chen, C. Y., Han, F., Wu, S. X., Tian, L., ... & Lu, T. X. (2015). Prognostic value of cervical nodal necrosis in nasopharyngeal carcinoma: analysis of 1800 patients with positive cervical nodal metastasis at MR imaging. Radiology, 276(2), 536-544.
- Feng, Y., Cao, C., Hu, Q., & Chen, X. (2019). Prognostic value and staging classification of lymph nodal necrosis in nasopharyngeal carcinoma after intensity-modulated radiotherapy. Cancer Research and Treatment: Official Journal of Korean Cancer Association, 51(3), 1222-1230.
- Liang, S. B., Chen, L. S., Yang, X. L., Chen, D. M., Wang, D. H., Cui, C. Y., ... & Xu, X. Y. (2021). Influence of tumor necrosis on treatment sensitivity and long-term survival in nasopharyngeal carcinoma. Radiotherapy and Oncology, 155, 219-225.
- Ou, D., Garberis, I., Adam, J., Blanchard, P., Nguyen, F., Levy, A., ... & Tao, Y. (2018). Prognostic value of tissue necrosis, hypoxia-related markers and correlation with HPV status in head and neck cancer patients treated with bio-or chemo-radiotherapy. Radiotherapy and Oncology, 126(1), 116-124.
- Zhang, X. C., Xu, C., Mitchell, R. M., Zhang, B., Zhao, D., Li, Y., ... & Zhao, L. P. (2013). Tumor evolution and intratumor heterogeneity of an oropharyngeal squamous cell carcinoma revealed by whole-genome sequencing. Neoplasia, 15(12), 1371-IN7.
- Lester, R. D., Jo, M., Montel, V., Takimoto, S., & Gonias, S. L. (2007). uPAR induces epithelial–mesenchymal transition in hypoxic breast cancer cells. The Journal of cell biology, 178(3), 425-436.
- Johansson, A. C., La Fleur, L., Melissaridou, S., & Roberg, K. (2016). The relationship between EMT, CD44high/EGFRlow phenotype, and treatment response in head and neck cancer cell lines. Journal of Oral Pathology & Medicine, 45(9), 640-646.